# Attitudes and practices of open data, preprinting, and peer-review—A cross sectional study on Croatian scientists

Ksenija Baždarić[1]*, Iva Vrkić[2], Evgenia Arh[3], Martina Mavrinac[1], Maja Gligora Marković[1], Lidija Bilić-Zulle[1], Jadranka Stojanovski[4,5], Mario Malički[6]

**1** Department of Medical Informatics, Faculty of Medicine, University of Rijeka, Rijeka, Croatia, **2** Department of Geophysics, Faculty of Science, University of Zagreb, Zagreb, Croatia, **3** Library, Faculty of Medicine, University of Rijeka, Rijeka, Croatia, **4** Department of Information Sciences, University of Zadar, Zadar, Croatia, **5** Centre for Scientific Information, The Ruđer Bošković Institute, Zagreb, Croatia, **6** Meta-Research Innovation Center at Stanford (METRICS), Stanford University, Stanford, California, United States of America

* ksenija.bazdaric@uniri.hr

## Abstract

Attitudes towards open peer review, open data and use of preprints influence scientists' engagement with those practices. Yet there is a lack of validated questionnaires that measure these attitudes. The goal of our study was to construct and validate such a questionnaire and use it to assess attitudes of Croatian scientists. We first developed a 21-item questionnaire called *Attitudes towards Open data sharing*, *preprinting*, *and peer-review* (ATOPP), which had a reliable four-factor structure, and measured attitudes towards open data, preprint servers, open peer-review and open peer-review in small scientific communities. We then used the ATOPP to explore attitudes of Croatian scientists (n = 541) towards these topics, and to assess the association of their attitudes with their open science practices and demographic information. Overall, Croatian scientists' attitudes towards these topics were generally neutral, with a median (Md) score of 3.3 out of max 5 on the scale score. We also found no gender (P = 0.995) or field differences (P = 0.523) in their attitudes. However, attitudes of scientist who previously engaged in open peer-review or preprinting were higher than of scientists that did not (Md 3.5 vs. 3.3, P<0.001, and Md 3.6 vs 3.3, P<0.001, respectively). Further research is needed to determine optimal ways of increasing scientists' attitudes and their open science practices.

## Introduction

Open science, despite lacking an universally accepted definition, is widely recognized as a global phenomenon and an initiative emerging from the philosophical concept of scholarly „openness". With the principles and values of openness rooted in the idea of scientific knowledge being a common good [1]. The term open science was coined in 2001 by Recep Şentürk, and he used it to refer to a democratic and a pluralist culture of science. For Şentürk, open science indicated that different perspectives in science are considered equal, rather than

**Data Availability Statement:** All relevant data are within the paper and Supporting information files.

**Funding:** The research was funded by the project Knowledge, attitudes and use of open science tools

in biomedicine (uniri-biomed-18-99) of the
University of Rijeka, Croatia.

**Competing interests:** The authors have declared
that no competing interests exist.

alternative to each other: "If we desire to recognize the complexity of our world we must embrace multiplex ontology" [1]. His view, however, is different from today's relatively narrow view of open science perceived as an „effort by researchers, governments, research funding agencies or the scientific community itself to make the primary outputs of publicly funded research results—publications and the research data—publicly accessible in digital format with no or minimal restriction"[2]. A recent systematic review summarized definitions of open science from 75 studies into „transparent and accessible knowledge that is shared and developed through collaborative networks"[3].

The open science movement intensified since 2010, when it became clear, that open access alone would not solve problems of non-reproducibility of published studies and the inaccessibility of research data, study protocols, laboratory notes, software, or peer review reports. The movement therefore also serves as a reminder of the basic tenets of science, and encourages open sciences to take the forefront in scholarly discussions [4].

Practical considerations of open science often deal with methods to lower or erase technical, social, and cultural barriers, and enable public sharing of all aspects of research [5], which are believed to lead toward the betterment of science [6]. Often, those practical considerations are described in various open science taxonomies and classifications, of which one of the most commonly used is the FOSTER's graphical representation, which distinguishes six „first level"elements of open science: open access, open data, open reproducible research, open science evaluation, open science policies, and open science tools [7].

In our research, we focused on the three of these elements: open data (open data use and reuse), open science tools (open repositories—preprint servers) and open science evaluation (open peer review).

## Open data

Open data are data that can be used (with proper attribution) by anyone without technical or legal restrictions [2]. Open Knowledge Foundation characterized them by: i) availability and access:; ii) reuse and re-distribution; iii) universal participation [8]. Many statements and recommendations were made to increase open data use and reuse [9], of which International Committee of Medical Journal Editors (ICMJE) recommendations, followed today by more than 500 biomedical journals, required a data sharing statement for clinical trials since July 2018 [10]. Research data is thought to be best preserved by being deposited in one of the many general or specific repositories existing today [11]. Although research and funding agencies often recognize the importance of data sharing, many technical and even psychological barriers still exist towards data sharing [12].

While the number of studies on open data has greatly risen in the last few decades [8], data is still rarely shared across sciences due to great differences between disciplines, debates on data ownership, lack of funding to support data sharing and data curation (i.e., preparation of data for sharing), as well as due to lack of incentives to reward it [9–15]. Recent estimates show that data sharing was mentioned in only of 15% biomedical [16], and in only 2% of psychological articles [17].

## Preprinting

Preprinting is an open science practice that allows the deposition and distribution of manuscripts (preprints) using an open science infrastructure (thematic or general preprint server) before submitting to a journal and being formally peer-reviewed [18–20]. While experiments with faster dissemination of research began in 1960s, in 1990s, first preprint servers (arXiv, SSRN and RePec) emerged and allowed public sharing of author's versions of manuscripts,

i.e., preprints, before those manuscripts were peer reviewed and published in journals (or other venues, such as books or conference proceedings). However, it took a while for preprint servers to become the go to place for researchers. For example, it took arXiv 8 years to become a major player in the dissemination of results in physics and mathematics [21]. Other scholarly fields have been even slower to adapt to the preprint culture: with bioRxiv, a preprint server dedicated to the biological sciences, originating in 2013, SocArXiv a server for preprints in social sciences in 2016, and MedRxiv, a server for clinical research preprints, in June of 2019 [19]. Further actualization and discussions surrounding preprint servers, also rose after Chalmers and Glasziou estimated that 85% of research is wasted due to inadequate research design or methodology, poor reporting, publication bias, and lack of scholarly openness [22], with some viewing preprint serves as a way to address some of these issues. Today, there are more than 60 preprint servers in the world covering all scholarly fields [23], and the number of preprints is rising, fuelled additionally by the COVID-19 pandemic [24]. Preprints are seen as a step toward greater openness of science, and in 2019, Fu and Hughey estimated that manuscripts first published as preprints received 36% more citations and had a 49% higher Altmetric score [25]. Increasing number of journals and funders today encourage preprinting [26]. Furthermore, many scholarly engines have started indexing preprints, e.g. Europe PMC [20], Scopus [18], and Dimensions [27].

## Open peer-review

Peer-review is a quality control mechanism for scholarly research or funding proposals. Traditionally, journal peer-review was most commonly blind (single, double or triple blind) and it was often criticised for being slow, expensive, subjective, not able to detect errors, non-reliable, prone to bias and easily abused [28]. This lead to growing need for a more open peer-review process [29]. Open peer-review as a term, however, lacks a universal definition [7]. Most often it used to describe one of the following practices: open identities of the authors and reviewers, open review reports published alongside the article, open interaction and discussion between author(s) and reviewers, or open platforms where a review is facilitated by a different entity than the one where the paper is published [7, 24, 30]. In our study, we consider open peer review to be open (public) sharing of review reports (with or without reviewers' names) as part of the journal or grant peer review processes. Practice and uptake of open peer review, however has been low, with less than 1% of journals today practicing it [31], and, to the best of our knowledge, no known estimates of its use by funders is available.

We are not aware of any studies, that analysed attitudes towards open data, preprinting and peer-review with a validated questionnaire. It was, therefore, our goal to construct and validate such a questionnaire; and use it to report attitudes towards open data, preprinting and open peer-review of Croatian scientists, as well as on the association between their attitudes and open science practices or their demographic information.

## Literature review

### Attitudes measurement

Attitudes can be defined as an individual's positive, neutral or negative feelings (evaluative affect) about a certain behaviour or a value [32, 33]. Attitudes are often measured with either one-item questions or with multi-item questionnaires (psychometric scales, whose answers are then often summarized to create a scale or an attitude score). While one-item questions can be a useful method for "snapshot measuring" [34], measuring attitudes with only a single question is generally not considered an optimal approach. On the other hand, creation of scales requires rigorous methodological approaches for questionnaire construction and validation [35–38].

This process often includes steps that include item/question generation, face validity checks, testing scales validity and reliability, and evaluating responsiveness and scale interpretability [35, 39–41]. A known fallacy of many attitude assessments is the difficulty to compare research findings, as the same questions or scales are rarely used multiple times or for different populations, and differences between studies can turn out to be a consequence of different wording of questions, emphasizing the need for creation of standardized questionnaires.

## Attitudes towards open data, preprinting and open peer-review

We present below our literature review of studies analysing attitudes towards open data, preprinting and open peer-review.

### Attitudes towards open data

In a recent (2020) systematic review, Zuiderwijk, Shinde and Jeng summarized results of 32 quantitative and qualitative studies on open data, of which 15 were surveys [42]. They found that "scholars refer to personal drivers and a positive attitude toward data sharing as vital individual drivers for openly sharing research data" and that they see a negative attitude as an inhibitor of data sharing [42]. In those summarized studies participants were mostly from the United States and Europe, and only a small number of studies were focused on multiple scientific disciplines. Most participants also had generally positive attitudes towards open data. An interesting finding was that in half of those studies (which assessed data sharing), there was no reference to studies own data availability [42].

An earlier 1988 study by Ceci described attitudes of 790 researchers from three US universities using a case scenario approach followed by 3 (snap-shot) questions, finding that researchers have a positive attitude towards data sharing, but acknowledging that those attitudes might have been influenced by giving socially desirable answers [43].

Two large studies of data sharing practices, and barriers of data reuse were authored by Tenopir et al. (2011 and 2015) using a multi-question approach, but without validating an attitude scale or reporting its reliability [44,45]. In the first study they surveyed approximately 1200 scientists, of which 900 were followed up in the second study. Most scientists were from North America (68%), and from the fields of environmental sciences and ecology (32%). Their results showed an increase in data sharing attitudes over time, but also an increase in the number of perceived barriers for data sharing [45]. Building on their questions, Curty et al. [13] later validated a scale for measuring attitudes towards data reuse on a sample of 570 scientist. They tested construct validity, and reported a 3 factor construct of their scale: perceived efficiency of data reuse (5 items), perception of data re-use (2 items) and concern about trustworthiness of data (4 items), with subscale reliability scores (Cronbach's alpha) ranging from 0.73 to 0.81 [13].

Yoon and Kim, in 2017, constructed and validated a scale using structural equation modelling (a combination of factor analysis and multiple regression), on a sample of 292 social scientists. Their questionnaire had 20 items and 7 factors (with Cronbach's alpha values ranging from 0.76 to 0.97). They also concluded that attitudes towards data reuse was a strong predictor of data reuse intention [46].

Zenk-Möltgen et al., in 2018, investigated attitudes towards data sharing of 446 political and sociology scientists using a theory of planned behaviour, but they did not report on their scale's validity or reliability. Overall, they found generally positive attitudes toward data and code sharing, and a strong association between previous sharing behaviour and intention to share [47].

Abele-Brehm et al., in 2019, investigated attitudes towards open data and data sharing of 337 psychological society members and reported a 2 factor scale (positive expectations—10 items with Cronbach's alpha of 0.90, and negative expectations—4 items with Cronbach's alpha of 0.67). They found that respondents attitudes were generally positive [48].

Finally, Zhu [12] in 2020, measured attitude of UK researchers towards data reuse with one question ("How important do you think it is, in general, to make research data available online for reuse?") and 1459 out of 1695 (86%) respondents found it to be very or fairly important.

## Attitudes towards preprinting

Studies evaluating attitudes towards preprinting are very scarce. Zha, Li and Yan, in 2013, have measured attitudes of 260 participants from natural and social sciences that previously posted a preprint on a Chinese preprint server [49]. Their questionnaire had 25 questions, with 7 factors (each construct had 2–5 items with Cronbach's alpha values from 0.85 to 0.98 with very high correlations indicating unidimensionality) and they found overall positive attitudes toward preprinting. Yi and Huh [50], investigated attitudes towards preprinting of 365 Korean authors and editors with 5 questions with a reliability of Cronbach α = 0.86, but did not report on the construct validity. Overall, they reported positive attitudes of respondents [50].

## Attitudes towards open peer-review

Twenty years ago, in 2001, a study by Melero and Lopez-Santovena, found that 17% of 103 reviewers for the journal *Food Science and Technology International* expressed favour fully open peer review (by answering a single question: "What system are you in favour of? Open or blinded") [51]. Ten years after that, 28% (104 out of 364) of Danish general medical journal reviewers expressed their preference for an open review (answering a single question: "Which peer review system do you prefer in the future?") [52]. One of the largest ever studies of attitudes towards open peer review was published by Ross-Hellauer, Deppe and Shmidt in 2017 [53], and although they used multiple questions, they did not report on their questionnaires validity or reliability. In total they collected approximately 3000 responses, mostly of researchers from Europe (61%), and from science, technology and medical (STM) fields (90%). Overall, respondents reported generally positive attitudes towards open peer-review. In 2018, Segado-Boj, Martín-Quevedo and Prieto-Gutiérrez, surveyed authors of Spanish journals (n = 295), mostly from social sciences (63%) with 7 questions, but they did not report on the questionnaires validity or reliability [54]. Overall, participants were found to be cautious towards open peer review. Lastly, in 2020, Besacon et al., using a small sample (N = 30) of researchers in the computer science field, and eight questions, reported that more than half of the respondents were in favour of open peer review, but not of displaying their reviewer names. They did not report the constructs validity or reliability [55].

## Materials & methods

We conducted a cross-sectional study with psychometrical validation of a questionnaire, which we named, the Attitudes towards Open data sharing, preprinting, and peer-review (ATOPP).

## Participants

In 2018, Croatia had 17,706 scientists [56]. In order to reach as most of them as we could, we sent invitations through 2 different channels: through the mailing list of Croatian scientists

(approximately 17,000 members) compiled by the Rudjer Boskovic Institute (Zagreb, Croatia), and the Dean's secretaries of University of Rijeka (the University of the first author, with 1,256 scientists).

## Procedure

Participants were invited to fulfil an anonymous online questionnaire (through Google forms). The survey was open from 12 May 2020 to 7 July 2020, and we sent two reminders 14 days apart.

**Constructing the questionnaire.** The questionnaire was constructed as a result of three focus groups we held at the University of Rijeka in 2019 and 2020 with a total of 24 participants. The first focus group was held with participants from Biomedical Sceinces (N = 12), second with the participants from Social Sciences (N = 7) and the last with participants from Natural Sciences (N = 5). Participants were asked 5 questions: (1) What is open science to you? (2) What are your experiences with open access journals? (3) What do you think about the open peer-review process? (4) Do you use any of the open science tools? (5) What could influence you to provide access to your research/project data? The sessions were recorded and the transcripts used for generating the survey questions [57]. The questionnaire face validity was then checked by us (the authors). This questionnaire had 73 questions, of which 45 were meant to assess the attitudes towards open science, specifically open access (8 items), open peer-review (12 items), open data (10 items), preprints (9 items), and open science tools (6 items). It also had 20 questions on open science practices; and 8 about demographic information. Answers to attitude statements were offered on a five-point Likert-type scale, where 1 indicated "strongly disagree;" 2 –"disagree;" 3 –"neither agree nor disagree;" 4 –"agree;" and 5 –"strongly agree." Open science practices questions were of mixed type (yes/no and multiple-choice questions). Demographic questions included questions on gender, age, scientific filed, roles in science, and the total number of published papers.

Our initial exploration (factor analysis) of the 45 attitudes questions showed that questions on open access (8 items) and open science tools (6 items) explained less than 5% of the variance of the total score and were not internally consistent (with Cronbach alpha scores <0.65) [35]. We then re-examined them (face validity), and hypothesized this is most likely due to the fact that these two aspects of open science dealt with concepts outside of direct researcher's influence (i.e. they were built by other actors), while data sharing, open peer review, and self-archiving through preprints were under direct (self-) agency of the researchers. The psychometrical validation of the remaining questions (31 items) is presented in the results.

## Statistical analysis

**Validation of the ATOPP questionnaire.** Construct validity of the scale was tested with exploratory factor analysis after the suitability of the item correlation matrix was checked with the Kaiser-Meyer-Olkin (KMO) measure of sampling adequacy and Bartlett's test of sphericity. In Exploratory Factor Analysis, we used Principal Axis Factoring (PAF) as the factor extraction method and Oblimin as the rotation method. We included the extracted factors with the eigenvalue >1, more than 5% of the construct variance and those which passed visual inspection on the scree plot. Factor loadings <0.30 are not presented [35]. The factor analysis procedure uses the pattern of correlation between questionnaire items, which represent directly measured manifest variables, grouping them by the variance they share which is captured by factors that are interpreted as latent dimensions, inferred constructs that are not directly measured. Consequently, each extracted factor or dimension is defined only by questionnaire items to which it

relates [35, 58]. Correlations of factors were calculated with Pearson's coefficient of correlation.

Internal consistency of the scale and subscales were determined with Cronbach alpha.

**Total score.** Before calculating the total score we have recorded 4 items: item 6 and 8 in Open data and items 10 and 11 in Open peer-review (S1 Appendix). The total score of whole scale and factors were constructed as a linear composite of all items divided by the 21 (number of items) with the score range being from 1 to 5. Lower results (<2.6) were considered as negative attitude, average (2.6–3.39) as neutral attitude and higher results (>3.39) as positive.

*Analysis of answers, based on the ATOPP survey*. Qualitative data are presented with frequency and relative frequency. Comparison of qualitative data is done with χ2 test and test of proportion.

Quantitative data are presented with median and interquartile range [Md(IQR)] and the distribution was tested with Kolmogorov- Smirnov test. Comparison of quantitative data was made with non-parametric (Mann-Whitney or Kruskal-Wallis) tests. Post-hoc test for Kruskal- Wallis was Dunn test.

For the purpose of the attitude analysis we have merged Natural sciences and Technical sciences, Biomedicine and health and Biotechnical sciences, and finally Social Sciences, Humanities and Interdisciplinary fields of science.

For statistical analysis, we have used 2 statistical packages SPSS (IBM SPSS Statistics for Windows, Version 25.0. Armonk, NY: IBM Corp) and Medcalc (MedCalc Software, Ostend, Belgium, version 16.0.3). P<0.05 was considered significant.

*Sample size calculation*. We based our calculation on the number of initial survey attitude questions (n = 45) and the fact that it is considered sufficient for scale validation to have 10 times more participants than the number of items [39].

## Ethics

The study was approved by the Ethical committee of the University of Rijeka, Rijeka, Croatia (KLASA: 003-08/19-01/l; URBROJ: 217 0-24-04-3–19–7). In the invite letter we also presented the informed consent form, which the participants had to approve in the online form before starting to fulfil the questionnaire.

## Results

### Validation of the ATOPP questionnaire

Thirty-one item related to open peer-review (12 items), open data (10 items) and preprinting (9 items) were entered into the exploratory factor analysis after exclusion of 14 items related to open access and open science tools (see Methods above). Acceptability of the construct was then assessed by analysing the floor and ceiling effects of the individual items on the score distribution and no floor and ceiling effects were observed.

Kaiser-Mayer Olkin test (KMO = 0.79) and the Bartlett's test of sphericity (P < 0.001) have satisfied the condition for principal axis factoring (PAF) of the 31 ATOPP questionnaire item. The inspection of the scree plot, Eigenvalues >1 and more than 5% of variance explained yielded 4 factors with 40% of the construct variance explained. We then repeated the factor analysis with 22 items (S1 Fig) that had factor loadings higher than 0.30 [35]. The second PAF analysis was more suitable (KMO = 0.80; Bartlett's test P<0.001) and it resulted with 4 factors (21 items)–Open Data, Preprinting, Open peer review in small scientific communities, and Open Peer-review that accounted with 51% of the construct variance (Table 1).

Structure matrix (correlations of each item with the extracted dimensions) is presented in S1 Appendix—Table 1, indicating 21 items were left in the model with a simple factorial

**Table 1. Attitudes towards open data, preprinting, and peer-review (ATOPP)—Reliability, factor loadings and median values.**

| Variable | Item factor loadings for Subscale* | | | | Median (IQR) |
|---|---|---|---|---|---|
| | Open data | Preprinting | Open peer review in small scientific communities | Open peer review | |
| **Cronbach α** | 0.80 | 0.82 | 0.85 | 0.73 | - |
| **Open peer review** | | | | | |
| 1. All journals should publish reviewers 'comments with reviewers' names. | | | | 0.737 | 2 (1–3) |
| 2. I would like to know who reviewed my work. | | | | 0.598 | 3 (2–4) |
| 3. If I have the opportunity to sign a review report I will always sign it. | | | | 0.472 | 3 (3–5) |
| 4. Reviews of papers that have been rejected should be available to all journals so that reviewers do not repeat the work. | | | | 0.423 | 3 (2–4) |
| 5. An open review of project proposals increases the transparency of the project selection process for funding. | | | | 0.443 | 4 (3–5) |
| 6. All public calls for projects should publish reviewers 'comments with the names of the reviewers. | | | | 0.683 | 3 (2–4) |
| 7. Smaller scientific communities should have a double-blind review of projects. | | | 0.897 | | 2 (1–3) |
| 8. Smaller scientific communities should have a double-blind review of papers in journals. | | | 0.804 | | 2 (1–3) |
| **Open data** | | | | | |
| 1. Data from scientific research should be publicly available. | 0.775 | | | | 5 (4–5) |
| 2. All collected (anonymous) research data financed by public funds should be public / open. | 0.739 | | | | 5 (4–5) |
| 3. All collected (anonymous) research data, regardless of the source of funding, should be public / open. | 0.677 | | | | 4 (3–5) |
| 4. I do not want my data to be downloaded and reused for other research. | -0.491 | | | | 4(3–5) |
| 5. If all or most of the data were publicly available, science would evolve faster. | 0.630 | | | | 4 (3–5) |
| 6. Authors should be able to decide who to give access to their research data. | -0.455 | | | | 3 (2–4) |
| 7. Journals should have access to all information during the review process. | 0.517 | | | | 4 (3–5) |
| 8. Each institution should have a repository for all data collected in its research. | 0.453 | | | | 4 (3–5) |
| **Preprinting** | | | | | |
| 1. Before sending the manuscript to the journal, I would publish the manuscript on a preprint server. | | 0.647 | | | 3 (2–4) |
| 2. Preprint servers can serve editors to select good manuscripts for their journal. | | 0.668 | | | 3 (3–4) |
| 3. Papers published in the preprint version achieve better citations than other papers. | | 0.755 | | | 3 (3–3) |
| 4. Papers published on preprint servers contribute to better visibility. | | 0.770 | | | 3 (3–4) |
| 5. By publishing the paper on the preprint server before sending it to the journal, I protect my work from a lengthy review process. | | 0.651 | | | 3 (2–3) |

*factor loadings—correlations with the total score in factor analysis; Recoded: Items 7 and 8 in Open peer—review and items 4 and 6 in Open data.

structure (loadings are distributed on one factor exclusively). The reliability of the whole scale was very good (Cronbach's alpha of 0.815).

## Participants' characteristics

We have collected 546 responses, 196 (36%) from University in Rijeka and 350 (64%) from the Rudjer Boskovic Institute list of Croatian scientists. There was no overlap between the respondents of the two sources, and 5 responses were not valid (not completed), leaving a total of 541

responses. The response rate for the University of Rijeka was 15.6% and for the Croatian scientist list it was 2%. Factorial structure of the attitude scales was the same for both samples and therefore we present them together.

Median age of the participants was 45 (38 to 53), with equal percentage of both males and females (43% vs 54%, P = 0.082) Majority of the respondents were from Biomedicine and Health (26%), Social (25%) or Natural Sciences (17%). They were most commonly Assistant Professors (29%), Full Professors (27%) or Associate Professor (19%). Most respondents (n = 529, 98%) published at least one article, with a median of 23 (IQR 10–45). More than two thirds (n = 371, 69%) were also reviewers, 16% (n = 87) acted as reviewers for funding agencies, and 11% (n = 62) as members of the editorial board, finally 3% (n = 18) were editors. Detailed demographic and scholarly information of respondents is presented in Table 2.

## Open science practices

Respondents' open science practices are presented in Table 3. Around half (47%, n = 240) of the respondents participated in open peer-review and most of them were happy to sign the review reports (n = 225, 95%).

Nearly half of the authors (46%, n = 249) published a paper in a journal in which research data could be deposited, and one third (29.9%, n = 162) published an article based on public data from other researchers. Most respondents shared their data (as supplementary files) via journals (54%, n = 285). Minority of the respondents posted a preprint (12%, n = 64), mostly on Arxiv (n = 38), BiorXiv (n = 12) or SocarXiv (n = 4).

## Attitudes towards open data, preprinting, and peer-review

The total score for all participants on the ATOPP scale was neutral with median of 3.3 (3.0–3.7). The neutral score was also found for their attitudes towards preprinting [3.0 (2.6–3.4)] and open peer review [3.2 (2.7–3.7)]. Negative attitude was found for the open peer-review in small scientific communities [2.0 (1.0–3.0)] and positive for open data [3.9 (3.4–4.4)] (all P<0.05) (Table 4). Differences in attitudes were tested regarding gender, field, open science practices and education (Table 4).

We found no gender differences (all P>0.05) except for the open peer-review in the small scientific communities, where female respondents had a more negative attitude than male respondents [2.0(1.0–3.0) vs 2.0(2.0–3.0), P = 0.032].

We also found no differences in the overall ATOPP score between scientific fields (P = 0.523). However, attitudes toward open peer review in small scientific communities were higher in Natural sciences and Technical sciences than in Social Sciences, Humanities and Interdisciplinary fields [2.5 (2.0–3.0) vs 2.0 (1.0–3.0), P = 0.002]. While attitudes towards open peer review were higher in Biomedicine and Health and Biotechnical sciences compared to Natural sciences and Technical sciences [3.3 (2.8–3.8) vs 3.0 (2.3–3.8), P = 0.023].

Participants who had open peer review experience had higher total ATOPP score (P<0.001), as well as attitudes towards open data (P = 0.008) and open peer-review (P<0.001). Similarly, those who previously shared their data had higher attitudes towards Open data (P = 0.007), Preprinting (P = 0.005) and Open peer review in small scientific communities (P = 0.021). Participants with experience in preprinting had more positive attitudes for all sub-scales (all P<0.05) except for the Open peer review in small scientific communities (P = 0.140). Finally, participants who had education in open science had a more positive ATOPP score then those that did not (<0.001) and they also had higher attitudes for preprinting and open peer review.

**Table 2. Study participants characteristics (n = 541).**

| Variable | n(%) |
|---|---|
| **Sex (n = 539)** | |
| Female | 290(54) |
| Male | 231(43) |
| Not declared | 18(3) |
| **Age (years) (n = 541)** | |
| <35 | 68 (13) |
| 35–44 | 196 (36) |
| 45–54 | 160 (30) |
| 55–64 | 88 (16) |
| >65 | 29 (5) |
| **Scientific field (n = 541)** | |
| Natural sciences | 94 (17) |
| Technical sciences | 67 (12) |
| Biomedicine and health | 140 (26) |
| Biotechnical sciences | 44 (8) |
| Social Sciences | 137 (25) |
| Humanities | 38 (7) |
| Interdisciplinary fields of science | 21 (4) |
| **Position in academia/science (n = 538)** | |
| Research Fellow | 35 (7) |
| Post Doc researcher | 47 (9) |
| Assistant Professor/Scientific associate | 156 (29) |
| Associate professor/Higher scientific associate | 105 (20) |
| Full professor/Scientific advisor | 148 (28) |
| Other | 47 (9) |
| **Published an article in a scientific journal (n = 541)** | |
| Yes | 529 (98) |
| No | 12 (2) |
| **Role†** | |
| Project associate | 423 (78) |
| Reviewer in a scientific journal | 371 (69) |
| Project manager | 204 (38) |
| Reviewer of scientific projects | 87 (16) |
| Member of the editorial board of a scientific journal | 62 (11) |
| Researcher in the industry | 19 (3) |
| Editor of a scientific journal | 18 (3) |
| Faculty management | 18 (3) |

*due to rounding, percentages don't always sum up to 100;

† Respondents could choose more than one role.

## Discussion

In this study we developed the ATOPP questionnaire for measuring attitudes toward open data, preprinting and open peer-review. Using the ATOPP questionnaire, we then explored Croatian scientists' attitudes towards those topics and the association of those attitudes with their open science practices and socio-demographic information. To the best of our knowledge, this is the first psychometrically validated (multiple-item) questionnaire for measuring

**Table 3. Open peer review, open data and preprinting practices.**

| Open science practice | n (%) |
|---|---|
| Reviewer allowed peer-review alongside the article (N = 525) | |
| Yes | 240 (46) |
| No | 285 (54) |
| Reviewer allowed open identity (N = 519) | |
| Yes | 225 (43) |
| No | 294 (57) |
| Author has published a journal article in which research data was available (N = 541) | |
| Yes | 249 (46) |
| No | 292 (54) |
| Author has published a journal article based on public data from other researchers (N = 541) | |
| Yes | 162 (30) |
| No | 379 (70) |
| Author posted a manuscript on a preprint server (N = 539) | |
| Yes | 64 (12) |
| No | 475 (88) |
| Preprint servers where authors archive | |
| ArXiv | 38 |
| BioRxiv | 12 |
| SocArXiv | 4 |
| PsyArXiv | 3 |
| ResearchGate | 3 |
| SSRN—Social Science Research Network Repository | 3 |
| Institutional repository | 2 |
| Academia.edu | 1 |
| Zenodo repository | 1 |
| ChemRxiv | 1 |
| Preprints.org | 1 |
| Education in open science (N = 474) | |
| Yes | 102 (22) |
| No | 372 (88) |

attitudes towards all these three topics with one questionnaire. The ATOPP scale, consisting of 21 items, demonstrated good internal consistency and validity. Because of its good psychometrical characteristics and relatively small number of questions, we believe that it represents a fast measurement that can be used in assessing or monitoring attitudes towards open science, thus allowing cross-cultural validation.

During ATOPP development, attitudes towards open peer review in small scientific communities turned out to be a separate factor (subscale) from attitudes toward open peer review. This could be a product of both the fact that Croatian scientific community for centuries had a higher number of specialized journals per capita compared to its neighbouring countries, and the fact that open peer review in small (national) fields or subfields has higher likelihood of reviewers being direct competitors for funding or job positions [58]. Additionally, smaller communities may experience greater fear of negative consequences of open peer review, i.e., fear of a potential (vindictive) backlash of their colleagues if they criticize their work, or if due to their review, they negatively affected funding or publication opportunities of their

**Table 4. Attitude towards open data, preprinting, and peer-review (ATOPP) of Croatian scientists (N = 541).**

| Variable | ATOPP scale score (Md, IQR) | Subscale score [Median (IQR)] | | | |
|---|---|---|---|---|---|
| | | Open data | Preprinting | Open peer review in small scientific communities | Open peer review |
| **Total score (N = 541)** | 3.3 (3.0–3.7) | 3.9 (3.4–4.4) | 3.0 (2.6–3.4) | 2.0 (1.0–3.0) | 3.2 (2.7–3.7) |
| **Gender** | | | | | |
| Female (n = 291) | 3.3 (3.0–3.7) | 3.9 (3.3–4.4) | 3.0 (2.6–3.6) | 2.0 (1.0–3.0) | 3.2 (2.7–3.7) |
| Male (n = 231) | 3.3 (3.0–3.6) | 4.0 (3.6–4.4) | 3.0 (2.4–3.4) | 2.0 (2.0–3.0) | 3.0 (2.5–3.7) |
| $P^*$ | 0.995 | 0.084 | 0.233 | 0.032 | 0.178 |
| **Field** | | | | | |
| Natural and Technical sciences (n = 161) | 3.3 (2.9–3.7) | 3.9 (3.4–4.4) | 3.0 (2.6–3.4) | 2.5 (2.0–3.0) | 3.0 (2.3–3.8) |
| Biomedicine and health and Biotechnical sciences (n = 184) | 3.3 (3.0–3.6) | 3.9 (3.4–4.3) | 3.0 (2.6–3.4) | 2.0 (1.5–3.0) | 3.3 (2.8–3.8) |
| Social Sciences, Humanities and Interdisciplinary (n = 196) | 3.3 (3.0–3.6) | 4.0 (3.3–4.6) | 3.0 (2.8–3.6) | 2.0 (1.0–3.0) | 3.2 (2.7–3.7) |
| $P$ | 0.523 | 0.088 | 0.123 | **0.002** | **0.023**§ |
| **Signed an open peer review report** | | | | | |
| YES (n = 225) | 3.5 (3.0–3.8) | 4.0 (3.6–4.6) | 3.0 (2.6–3.6) | 2.0 (1.0–3.0) | 3.3 (2.8–4.0) |
| NO (n = 294) | 3.3 (2.9–3.5) | 3.9 (3.3–4.3) | 3.0 (2.1–3.6) | 2.0 (1.0–3.0) | 3.0 (2.5–3.5) |
| $P^*$ | **<0.001** | **0.008** | 0.663 | 0.413 | **<0.001** |
| **Shared data for their study** | | | | | |
| YES (n = 249) | 3.3 (3.0–3.7) | 4.0 (3.4–4.6) | 3.0 (2.8–3.0) | 2.5 (1.9–3.0) | 3.2 (2.5–3.8) |
| NO (n = 292) | 3.3 (3.0–3.6) | 3.9 (3.3–4.3) | 3.0 (3.0–3.2) | 2.0 (1.0–3.0) | 3.2 (2.7–3.7) |
| $P^*$ | 0.520 | **0.007** | **0.005** | **0.021** | 0.722 |
| **Posted a preprint** | | | | | |
| YES (n = 64) | 3.6 (3.1–3.7) | 4.2 (3.5–4.6) | 3.6 (3.0–4.0) | 2.5 (1.0–3.5) | 3.0 (2.3–3.5) |
| NO (n = 475) | 3.3 (3.0–3.6) | 3.9 (3.4–4.4) | 3.0 (2.6–3.4) | 2.0 (1.0–3.0) | 3.2 (2.7–3.7) |
| $P^*$ | **0.006** | **0.017** | **<0.001** | 0.140 | **0.044** |
| **Participated in a course on open science** | | | | | |
| YES (n = 102) | 3.6 (3.2–3.7) | 4.0 (3.6–4.6) | 3.2 (2.8–3.8) | 2.0 (1.0–3.0) | 3.3 (2.8–3.8) |
| NO (n = 372) | 3.3 (3.0–3.6) | 3.9 (3.4–4.4) | 3.0 (2.6–3.4) | 2.0 (1.0–3.0) | 3.0 (2.5–3.7) |
| | **<0.001** | 0.076 | **<0.001** | 0.671 | **0.025** |

* Mann Whitney U test,

† Kruskal-Wallis test,

‡ Respondents from Natural and Technical sciences differed significantly from those of Social Sciences, Humanities and Interdisciplinary fields;

§- Respondents from Natural and Technical sciences differed significantly from Biomedicine and health and Biotechnical Sciences; Score interpretation: <2.6 –negative attitude, 2.6–3.39 –neutral attitude, >3.39 –positive attitude.

colleagues. And these fears will likely remain until external evaluations or strong protective mechanisms are put into practice (however, for small communities these approaches likely face significant language barriers and high costs).

Based on the ATOPP questionnaire, we then found that the Croatian scientists had generally neutral attitudes toward open science. Their most positive attitudes were towards open data, while their attitude towards preprinting and towards open peer-review were neutral, and those towards open peer review in smaller scientific communities were negative. We also found no gender or scholarly field differences in respondents' overall attitude scores. However, scientists who already had experience with open science practices, i.e., shared data, provided open peer review reports in the past, or posted preprints, had generally more positive attitudes

than those who did not. Higher attitude in those with experience in open science practices and previous open science education are in accordance with Bem's self-perception theory that confirms effect of past behavior on internal attitude [59]. Past behaviour influence on attitudes was also confirmed by many researchers since then [60, 61]. We have also found that participants who had taken open science courses had more positive ATOPP scale score, preprinting score and open peer-review score. which is a confirmation of a model that positive attitudes are related to intention to behave and behavior [32].

The positive attitudes towards open data in our study were associated with the high prevalence of researchers in our sample (46%) that shared their data in the past. However, in the recent survey by Zhu (2020) in the United Kingdom on 1724 participants from various scholarly fields, there were less participants (21%) who had deposited primary data in online repositories, although majority (86%) had a very positive attitude towards data sharing [12]. However, in that survey, attitude was only measured by a single question "How important do you think it is, in general, to make research data available online for reuse?".

Positive attitude towards open data in our study can be compared with the data in a recent survey among members of the German psychological society (N = 303). Abele-Brehm et al. constructed a scale measuring hopes (10 items, Cronbach $\alpha$ = 0.90) and fears (4 items, Cronbach $\alpha$ = 0.67) towards data sharing. The positive attitude–"hopes" of respondents were neutral, but their experience with data sharing was not measured [48]. Yoon and Kim (2017) investigated data reuse behaviour by measuring beliefs, attitudes, and norms. Based on their theoretical framework attitude was a strong positive predictor of data reuse [46].

Attitudes of Croatian scientist towards preprinting were neutral in our study, except of those scientists who preprinted in the past (12%). These results differ from attitudes participants in South Korea [50], China [49], and Latin America [62] whose attitudes were overall found to be positive; but those surveys included more editors, and all (China), or many respondents who previously posted a preprint (Korea (32% editors, 15% past preprint users; and Latin America, 40% past preprint users), while in our sample only 12% of scientists did so and we had only 11% of editorial board members in the sample [49, 50]. Additionally, China introduced a country preprint server ChinaXiv in 2006, and Latin America in 2020 (ScIELO preprints), which most likely further promoted already strong open access culture in those countries (Croatia does not have a national preprint server). Croatian scientist preprint use in our study, is in the line with a large analysis of Biorxiv preprints (n = 67,885, in the period from 2013 to 2019), which found that senior authors of preprints are often researchers from the United States (39.2%) and the United Kingdom (10.5%), while Croatia was described as a "contributor country" whose authors were rarely on senior authorship positions [63]. More studies are, however, needed to determine the main factors that drive researchers to start preprinting manuscripts or project proposals (protocols), as well as inviting or choosing to wait for public comments before deciding to submit the preprint for scholarly journal peer review. Additionally, previous survey has shown that scientist choices toward posting a preprint are influenced by the policies of the journals in which they plan to publish those studies [62, 64]. Although Croatia has approximately 400 active scholarly journals [65] of which less than half are indexed in WoS or Scopus [66], preprint policies are listed for only 24 in Sherpa website, and so further research is needed to determine the influence of Croatia's editorial, funder and publishing milieu on the preprint attitudes of its researchers.

In our study, Croatian scientists' overall attitudes towards open peer review were neutral. We also found differences among scientific fields with scientists from Biomedicine and Health and Biotechnical sciences having higher attitude score (albeit still neutral), as did those with previous experience with open peer review. In Ross-Hellauer, Deppe and Schmidt 2017 study on 3062 participants, most thought that open peer-review should be common practice, with

the strongest support coming from social science researchers (e.g. economics, psychology and philosophy). Also, the majority of researchers agreed that the obligatory signing of review reports is strongly associated with rejecting peer review requests [53]. That study however did not report on the validity and reliability of its questionnaire.

Attitude towards open peer-review in small scientific communities of Croatian scientist were low and even lower than attitudes towards open peer-review, indicating that Croatian scientific community might not be ready for this aspect of open science yet. Female scientists also had a lower attitude than male scientists, which could be the results of the gender disbalance in academia, and greater fear of retaliation or promotion obstruction [67]. Slightly higher attitudes towards open peer-review in small scientific communities was found among scientists who shared data before, preprinted, or were from natural and technical sciences; which are fields that have been sharing preprints the longest.

Despite presenting the first psychometrically validated scale for measuring attitudes towards open data, preprinting and open peer-review, our study is not without limitations. As all questionnaires, are data are based on self-declared attitudes and open science practices and does not capture independently confirmed practices. Furthermore, as in many recent online surveys, our response rates were low, and that rate might have also been influenced by the fact that the questionnaire was sent during the early months of the COVID-19 pandemic. Additionally, we might have captured opinions only of those interested in these topics, which, if true, could mean that the attitudes of a representative sample of Croatian scientists would be even lower towards these topics. While we did provide definitions of open science practices in our questionnaire, as most of our respondents (78%) did not have education in open science, it is possible some held different ideas of those practices. Finally, while our study, showed a strong association between open sciences attitudes and previous open science practices, and in that way provides further credibility to the ATOPP questionnaire, our study was cross-sectional and was not designed to look at the possible consequences of these attitudes on promotion and implementation of open science practices. Further interventional research is needed to explore the most efficient interventions in increasing positive researchers' attitudes toward open science practices, as well as if those interventions would lead to greater uptake of such practices [68]. Furthermore, with the recent changes in the EU funding schemes for the period of 2021 to 2027, and the requirement for open peer review and data sharing [69]; announcement of the journal eLife for only accepting submissions if they have been posted as preprints [70], as well as dedicated calls for research into ways to increase open sciences practices of those who have not embraced them yet [71] it will be interesting to compare if approaches aimed at rewarding open science practices vs mandating of those practices by several key stakeholders, will have a different impact on researchers attitudes. And, ultimately, which of those approaches will be more effective in inducing changes in the wider scholarly community.

## Conclusions

In conclusion, our study presents the validation of a multi-item questionnaire for measuring open science attitudes, specifically open data, preprinting and open peer review. Additionally, using the questionnaire we found that attitudes Croatian researchers towards these topics were neutral, and that more positive attitudes were found among those that participated in open science practices before or had an education in open science. Further studies are needed to assess attitudes of researchers on these topics in other countries, as well as to track changes of these attitudes over time. With more and more funders and institutions encouraging or mandating

open science practices, we believe that validated tools, such as this one, could help assess and monitor researchers' attitudes and their associations with open science practices.

## Supporting information

**S1 Appendix. Factor analysis of the attitudes and practices of open data, preprinting, and peer-review—a cross sectional study on Croatian scientists.**
(PDF)

**S2 Appendix. ATOPP questionnaire questions.**
(PDF)

**S1 Fig. Scree plot of the factor analysis (22 items) of attitudes towards open data, preprinting, and peer-review (ATOPP).**
(TIF)

**S1 Dataset.**
(XLSX)

## Author Contributions

**Conceptualization:** Ksenija Baždarić, Martina Mavrinac, Maja Gligora Marković, Lidija Bilić-Zulle, Jadranka Stojanovski, Mario Malički.

**Data curation:** Ksenija Baždarić, Iva Vrkić, Evgenia Arh, Mario Malički.

**Formal analysis:** Ksenija Baždarić, Iva Vrkić, Martina Mavrinac.

**Funding acquisition:** Ksenija Baždarić.

**Investigation:** Ksenija Baždarić, Evgenia Arh, Martina Mavrinac, Maja Gligora Marković, Jadranka Stojanovski, Mario Malički.

**Methodology:** Ksenija Baždarić, Iva Vrkić, Evgenia Arh, Martina Mavrinac, Maja Gligora Marković, Lidija Bilić-Zulle, Jadranka Stojanovski, Mario Malički.

**Project administration:** Ksenija Baždarić.

**Supervision:** Ksenija Baždarić, Mario Malički.

**Visualization:** Ksenija Baždarić.

**Writing – original draft:** Ksenija Baždarić, Iva Vrkić, Evgenia Arh, Mario Malički.

**Writing – review & editing:** Ksenija Baždarić, Iva Vrkić, Evgenia Arh, Martina Mavrinac, Maja Gligora Marković, Lidija Bilić-Zulle, Jadranka Stojanovski, Mario Malički.

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
