## [Decision Letter · Decision Letter 0]

25 Feb 2021

PONE-D-20-39499

Attitudes and Practices of Open Data, Preprinting, and Peer-review - a Cross Sectional Study on Croatian Scientists

PLOS ONE

Dear Dr. Bazdaric,

Thank you for submitting your manuscript to PLOS ONE. After careful consideration, we feel that it has merit but does not fully meet PLOS ONE’s publication criteria as it currently stands. Therefore, we invite you to submit a revised version of the manuscript that addresses the points raised during the review process.

As you can see below, the three reviewers found your work interesting and technically sound. However, they also highlighted some issues concerning the manuscript that should be addressed. In particular:

- Literature review: As stated by Reviewer 3, a solid literature review would help to frame your work and better identify the gaps you are covering.

- Discussion: All three reviewers requested a further development of the discussion section. They also provided some suggestions on how to do it.

We look forward to receiving your revised manuscript.

Kind regards,

Sergi Lozano

Academic Editor

PLOS ONE

Journal Requirements:

Reviewers' comments:

Reviewer's Responses to Questions

**Comments to the Author**

1. Is the manuscript technically sound, and do the data support the conclusions?

Reviewer #1: Yes

Reviewer #2: Yes

Reviewer #3: Partly

2. Has the statistical analysis been performed appropriately and rigorously? 

Reviewer #1: Yes

Reviewer #2: Yes

Reviewer #3: I Don't Know

3. Have the authors made all data underlying the findings in their manuscript fully available?

Reviewer #1: Yes

Reviewer #2: Yes

Reviewer #3: Yes

4. Is the manuscript presented in an intelligible fashion and written in standard English?

Reviewer #1: Yes

Reviewer #2: Yes

Reviewer #3: No

5. Review Comments to the Author

Reviewer #1: Excellent scale and method. This work is needed and great care was taken to address the validity of the survey. I think much more could be expanded upon in the discussion and conclusion. These implications go beyond what is introduced in this version as well as more detail on next steps of the tool and how else it may be used in future research. The impact will be broader if these sections are more organized and lengthier. I would personally like to know more about the open data attitudes since this was not discussed as much as open peer review.

Reviewer #2: This manuscript does a great job demonstrating the development of the ATOPP questionnaire for measuring attitudes toward open data, open peer review and preprints. "Open data" and "Data sharing " have been discussed for a long time, and they are still essential topics nowadays. This paper examines Croatian scientists’ attitudes towards not only open data, but also includes open peer review and preprints, which are significant to the academic community.

The authors perform effective statistical procedures to conduct this research. First, they validated the questionnaire by using Kaiser-Meyer-Olkin to measure sample adequacy and Bartlett's test of sphericity to text the suitability of the item correlation matrix. Then, they statistically analyzed the collected data to draw conclusions.

Finally, the authors found that the prevalent attitudes concerning preprint and open peer review are neutral; negative attitudes were found for the open peer-review in small scientific communities; and the attitude regarding open data is positive. The Result section is described concisely, and I do not have questions about the results. However, I would suggest the authors enrich the Discussion section. For instance, what are the implications of these attitudes? Or are there any solutions which can solve or relieve scientists’ concerns about the negative attitudes?

This article is well written; however, it misses an important section— a literature review. The authors provided concepts about the key terms, such as preprinting, open peer-review, and open data, but no related works on the research topics. Even though the authors claim that this work is the first psychometrically validated scale for measuring attitudes towards those topics using a multiple questions approach, there must be related scholarly work on scientists’ perspectives, ideas, or attitudes concerning open data and open science. I think the authors should examine those research studies. Based on my experience, almost all academic papers include the literature review/related works in the front or last part.

A small typo in line 381, I “belie” that… should be I “believe”?

Reviewer #3: There is a good start here and there is much within the data use/reuse area that needs clearer definitions and validated data collection methods.

I think this paper has the potential to help contribute to studying data practices and beliefs through their validated questionnaire.

The authors assert that the main goal of the study was the construction of the valid questionnaire.

However, there is no literature review to support that goal or in fact no literature review to speak of. Some effort is made to define various concepts that are not as they admit not well defined. I would expect a review of other studies whose main goal was the construction of a data collection instruments that attempt to measure beliefs in an emerging area.

Such a literature review, would then have been helpful in terms of evaluating the methods the authors to construct validity. As it stands, the reader must either come with this body of knowledge or trust their estimation. They have not made a case for the methods they’ve used to support their main goal.

Next, the discussion of the results lies in the assessment of the attitudes of the scientists and not the validity of the questionnaire. The authors need to connect the two aims of this paper. Clearly there are two aims: 1) the questionnaire and 2) the analysis of the questionnaire. Not the one as stated. The authors should describe how the two aims are interrelated and inform each other.

Lastly, the paper needs to be copy edited: punctuation and capitalization practices are not grammatically correct.

6. PLOS authors have the option to publish the peer review history of their article (what does this mean?). If published, this will include your full peer review and any attached files.

Reviewer #1: **Yes: **Bradley Bishop

Reviewer #2: No

Reviewer #3: No

---

## [Author Response · Author response to Decision Letter 0]

19 Apr 2021

Dear Editor Sergi Lozano, 

Thank you and the reviewers for the constructive suggestions to improve our manuscript. We have prolonged the deadline because I was sick so thank you for letting us work 2 more weeks on the manuscript. 

We believe we have addressed all suggestions appropriately, and we present them below in a point-by-point manner.

Editor’s comments:

#1 Literature review: As stated by Reviewer 3, a solid literature review would help to frame your work and better identify the gaps you are covering.

Reply: We have included a detailed literature review in the revised version for each of the 3 concepts we introduced in the introduction and expanded on their definitions and uptake. We would also like to mention to the editor, that we checked all data in the table and found a number that we wrongly copied, and so we corrected it (this does not in any way change any of our results – was just a misspell). 

#2 Discussion: All three reviewers requested a further development of the discussion section. They also provided some suggestions on how to do it.

Reply: We thank the reviewers for their suggestions and have rewritten most of the discussion. 

#3 Technical details:

• Please ensure that your manuscript meets PLOS ONE's style requirements, including those for file naming. Style templates can be found at link. 

Reply: All 3 files are appropriately named, and submitted with the revised version, and we followed PLOS ONE’ style requirements. 

#4 Please provide additional details regarding participant consent. In the ethics statement in the Methods and online submission information, please ensure that you have specified what type you obtained (for instance, written or verbal, and if verbal, how it was documented and witnessed). If your study included minors, state whether you obtained consent from parents or guardians. If the need for consent was waived by the ethics committee, please include this information.

Reply: The methods section and the online submission provide all information on the consent procedures during the online survey. We have indicated in the methods “In the invite letter we also presented the informed consent form, which the participants had to approve in the online form before starting to fulfil the questionnaire.” 

#5. Please amend your list of authors on the manuscript to ensure that each author is linked to an affiliation. Authors’ affiliations should reflect the institution where the work was done (if authors moved subsequently, you can also list the new affiliation stating “current affiliation:” as necessary).

Reply: All authors are now linked with their affiliations. 

Reviewers' comments:

Reviewer #1: 

#5 Excellent scale and method. This work is needed and great care was taken to address the validity of the survey. I think much more could be expanded upon in the discussion and conclusion. These implications go beyond what is introduced in this version as well as more detail on next steps of the tool and how else it may be used in future research. The impact will be broader if these sections are more organized and lengthier. I would personally like to know more about the open data attitudes since this was not discussed as much as open peer review.

Reply: We thank the reviewer for his kind words and suggestions about our manuscript. We have included a detailed literature review section in the manuscript, and greatly expanded the discussion. The literature review also now includes the information on open data attitudes, and the discussion includes considerations of recent developments and their possible influence on researchers’ attitudes.

Reviewer #2:

#6 This manuscript does a great job demonstrating the development of the ATOPP questionnaire for measuring attitudes toward open data, open peer review and preprints. "Open data" and "Data sharing " have been discussed for a long time, and they are still essential topics nowadays. This paper examines Croatian scientists' attitudes towards not only open data, but also includes open peer review and preprints, which are significant to the academic community.The authors perform effective statistical procedures to conduct this research. First, they validated the questionnaire by using Kaiser-Meyer-Olkin to measure sample adequacy and Bartlett's test of sphericity to text the suitability of the item correlation matrix. Then, they statistically analyzed the collected data to draw conclusions. Finally, the authors found that the prevalent attitudes concerning preprint and open peer review are neutral; negative attitudes were found for the open peer-review in small scientific communities; and the attitude regarding open data is positive. The Result section is described concisely, and I do not have questions about the results. However, I would suggest the authors enrich the Discussion section. For instance, what are the implications of these attitudes? Or are there any solutions which can solve or relieve scientists' concerns about the negative attitudes?

Reply: We thank the reviewer for his kind words. The revised manuscript includes a greatly expanded discussion, where we touch upon reasons behind the scores of these attitudes, as well as discuss the need for studies to see the interplay of attitudes and promotion and implementation of open science practices, as well as solutions to relive the negative attitudes. 

#7: This article is well written; however, it misses an important section— a literature review. The authors provided concepts about the key terms, such as preprinting, open peer-review, and open data, but no related works on the research topics. Even though the authors claim that this work is the first psychometrically validated scale for measuring attitudes towards those topics using a multiple questions approach, there must be related scholarly work on scientists' perspectives, ideas, or attitudes concerning open data and open science. I think the authors should examine those research studies. Based on my experience, almost all academic papers include the literature review/related works in the front or last part.

Reply: We have included a detailed literature review in the revised version of the manuscript.

#8: A small typo in line 381, I "belie" that... should be I "believe"?

Reply: We thank the reviewer for noticing this. We corrected it, and rephrased several other sentences to increase their clarity. 

Reviewer #3

#9: There is a good start here and there is much within the data use/reuse area that needs clearer definitions and validated data collection methods. I think this paper has the potential to help contribute to studying data practices and beliefs through their validated questionnaire.

The authors assert that the main goal of the study was the construction of the valid questionnaire. However, there is no literature review to support that goal or in fact no literature review to speak of. Some effort is made to define various concepts that are not as they admit not well defined. I would expect a review of other studies whose main goal was the construction of a data collection instruments that attempt to measure beliefs in an emerging area. Such a literature review, would then have been helpful in terms of evaluating the methods the authors to construct validity. As it stands, the reader must either come with this body of knowledge or trust their estimation. They have not made a case for the methods they’ve used to support their main goal.

Reply: We thank the reviewer for the kind words and have included a detailed literature review in the revied manuscript, alongside description and references on means of constructing and validating psychometric scales/questionnaires. We thank the reviewer especially for this last point, as while it might be standard in psychology field for all students to have taken courses on scale development, this might be less known to researchers from other fields. 

#10: Next, the discussion of the results lies in the assessment of the attitudes of the scientists and not the validity of the questionnaire. The authors need to connect the two aims of this paper. Clearly there are two aims: 1) the questionnaire and 2) the analysis of the questionnaire. Not the one as stated. The authors should describe how the two aims are interrelated and inform each other.

Reply: We thank the reviewer for pointing out this needed further clarification, and we have therefore emphasized these two goals in the revised manuscript, both in the introduction, abstract, and in the discussion section. 

Lastly, the paper needs to be copy edited: punctuation and capitalization practices are not grammatically correct.

Reply: We thank the reviewer for noticing these. We have made many changes to phrasing, punctuation, and capitalization in the manuscript, and believe we corrected all of them. 

We would like to thank the editor and the reviewers again for their comments, and hope that you will find the revised manuscript suitable for publication in your journal.

Kind regards, 

in the name of the co-authors

Ksenija Baždarić

---

## [Decision Letter · Decision Letter 1]

11 May 2021

Attitudes and Practices of Open Data, Preprinting, and Peer-review - a Cross Sectional Study on Croatian Scientists

PONE-D-20-39499R1

Dear Dr. Bazdaric,

We’re pleased to inform you that your manuscript has been judged scientifically suitable for publication and will be formally accepted for publication once it meets all outstanding technical requirements.

Kind regards,

Sergi Lozano

Academic Editor

PLOS ONE

Additional Editor Comments (optional):

Reviewers' comments:

Reviewer's Responses to Questions

**Comments to the Author**

1. If the authors have adequately addressed your comments raised in a previous round of review and you feel that this manuscript is now acceptable for publication, you may indicate that here to bypass the “Comments to the Author” section, enter your conflict of interest statement in the “Confidential to Editor” section, and submit your "Accept" recommendation.

Reviewer #2: All comments have been addressed

Reviewer #3: All comments have been addressed

2. Is the manuscript technically sound, and do the data support the conclusions?

Reviewer #2: Yes

Reviewer #3: Yes

3. Has the statistical analysis been performed appropriately and rigorously? 

Reviewer #2: Yes

Reviewer #3: Yes

4. Have the authors made all data underlying the findings in their manuscript fully available?

Reviewer #2: Yes

Reviewer #3: Yes

5. Is the manuscript presented in an intelligible fashion and written in standard English?

Reviewer #2: Yes

Reviewer #3: Yes

6. Review Comments to the Author

Reviewer #2: This new manuscript adequately addressed two of my comments raised in the first round of review, including adding a Literature Review section and enriching the Discussion section. Therefore, I think that this manuscript is now acceptable for publication.

Reviewer #3: This revision contains a solid literature review and a discussion section that contextualizes and makes clear the contribution to open science this paper is making.

7. PLOS authors have the option to publish the peer review history of their article (what does this mean?). If published, this will include your full peer review and any attached files.

Reviewer #2: No

Reviewer #3: No

---

## [Editor Report · Acceptance letter]

9 Jun 2021

PONE-D-20-39499R1 

Attitudes and Practices of Open Data, Preprinting, and Peer-review - a Cross Sectional Study on Croatian Scientists 

Dear Dr. Baždarić:

I'm pleased to inform you that your manuscript has been deemed suitable for publication in PLOS ONE. Congratulations! Your manuscript is now with our production department. 

Kind regards, 

on behalf of

Dr. Sergi Lozano 

Academic Editor

PLOS ONE